# Stopping Feline Coronavirus Shedding Prevented Feline Infectious Peritonitis

**DOI:** 10.3390/v15040818

**Published:** 2023-03-23

**Authors:** Diane D. Addie, Flora Bellini, Johanna Covell-Ritchie, Ben Crowe, Sheryl Curran, Mark Fosbery, Stuart Hills, Eric Johnson, Carrie Johnson, Steven Lloyd, Oswald Jarrett

**Affiliations:** 1Maison Zabal, 64470 Etchebar, France; 2Independent Researcher, Flora Bellini, Uxbridge, UK; 3Independent Researcher, Johanna Covell-Ritchie, Maidstone, UK; 4Independent Researcher, Ben Crowe, Uxbridge, UK; 5Independent Researcher, Sheryl Curran, Baker Street Ragdolls, Liverpool, UK; 6Mark Fosbery, Newnham Vets, Maidstone, UK; 7Stuart Hills, Ark Veterinary Centre, Lockerbie, UK; 8Independent Researcher, Eric Johnson, Firestone, CO, USA; 9Independent Researcher, Carrie Johnson, Firestone, CO, USA; 10Steve Lloyd, Zoologix Laboratory, CA, USA; info@zoologix.com; 11Oswald Jarrett, Glasgow, UK

**Keywords:** feline coronavirus, feline infectious peritonitis, FIP prevention, GS-441524, chronic enteritis, inflammatory bowel disease, IBD, antiviral, diarrhoea, itraconazole

## Abstract

After an incubation period of weeks to months, up to 14% of cats infected with feline coronavirus (FCoV) develop feline infectious peritonitis (FIP): a potentially lethal pyogranulomatous perivasculitis. The aim of this study was to find out if stopping FCoV faecal shedding with antivirals prevents FIP. Guardians of cats from which FCoV had been eliminated at least 6 months earlier were contacted to find out the outcome of their cats; 27 households were identified containing 147 cats. Thirteen cats were treated for FIP, 109 cats shed FCoV and 25 did not; a 4–7-day course of oral GS-441524 antiviral stopped faecal FCoV shedding. Follow-up was from 6 months to 3.5 years; 11 of 147 cats died, but none developed FIP. A previous field study of 820 FCoV-exposed cats was used as a retrospective control group; 37 of 820 cats developed FIP. The difference was statistically highly significant (*p* = 0.0062). Cats from eight households recovered from chronic FCoV enteropathy. Conclusions: the early treatment of FCoV-infected cats with oral antivirals prevented FIP. Nevertheless, should FCoV be re-introduced into a household, then FIP can result. Further work is required to establish the role of FCoV in the aetiology of feline inflammatory bowel disease.

## 1. Introduction

Coronaviruses are epitheliotropic single-strand positive-sense RNA viruses belonging to the order Nidovirales, family *Coronaviridae,* and feline coronavirus (FCoV) is a member of the genus *Alphacoronavirus.* The FCoV species is further divided into two types: I and II, the first type being wholly feline, and type II being a recombinant hybrid between type I and canine coronavirus (CCoV) [1,2]. Type-I FCoV is maintained in a group of cats by a cycle of infection, continuous virus shedding for some months in faeces, and then a cessation of virus shedding [3,4,5]. Immunity is short lived, so transient infection may be followed by re-infection by the same strain or a different coronavirus strain [5]. We previously believed that intermittent coronavirus shedding could occur [4] but now our view is that this belief was erroneous and that the appearance of intermittent virus shedding was due to re-infection (usually by infected cats in the same household); virus shedding at the limits of detection of the assay concerned (a nested reverse-transcriptase polymerase chain reaction (RT-PCR) test was used during our previous publication [4]); and/or the inhibition of PCR tests by faecal inhibitors [6] or cat litter. Around 13% of type-I FCoV-infected cats become persistently infected—i.e., carrier cats [4]—shedding the same strain of the virus in the faeces continuously [5]. In carrier cats, the virus replicates in the large intestine, especially the colon [7]. Whether or not type-II FCoV can induce a carrier status is unknown.

Clinically, FCoV is a major cause of acute diarrhoea [8] and is usually in kittens [9,10], but it can affect cats of any age and can occasionally be fatal [11]. Chronic FCoV infection has been reported to cause chronic diarrhoea [4]. FCoV fulfils at least three of the five criteria which are required by the World Small Animal Veterinary Association International Gastrointestinal Standardization Group for a diagnosis of inflammatory bowel disease (IBD) [12] also called chronic enteropathy (CE) [13]. The clinical criteria that chronic FCoV enteritis fulfils are first chronic (i.e., over 3 weeks in duration) gastrointestinal signs (e.g., anorexia, vomiting, weight loss, diarrhoea, hematochezia, and mucoid faeces); second, an inability to document other causes (e.g., parasites, protozoa, and bacteria) of gastroenterocolitis by thorough diagnostic evaluation; third, inadequate response to appropriately designed and implemented therapeutic trials (i.e., dietary, probiotic, antibacterial, and anthelmintic) [12,13,14,15,16]. Whether FCoV enteropathy fulfils the fourth criterion—histopathologic evidence of mucosal inflammation—is unknown, and the fifth criterion, a clinical response to anti-inflammatory or immunosuppressive agents, tends to be met only temporarily, if at all. The mortality rate of FCoV enteritis is unknown.

A minority of FCoV-infected cats develop a potentially lethal pyogranulomatous perivasculitis, feline infectious peritonitis (FIP) [17]. In a study of 420 naturally infected cats in 33 households where FIP occurred, mortality due to FIP was 14% at the time of the first FIP death(s), reducing to 8.8% subsequently [18]. FIP mortality in 282 FCoV antibody-positive kittens was 7.8% [18].

We reported previously that a 4−7-day course of Mutian Xraphconn pills stopped the faecal shedding of FCoV [19] (Mutian pills were subsequently shown to contain the adenosine nucleoside analogue GS-441524 [20].) We then considered whether this brief treatment prevented FIP; the outstanding question was whether the antiviral drug had simply eliminated the virus from the gut, or whether the virus persisted elsewhere in the body, allowing FIP to develop at a later date. In this paper, we provide evidence that early treatment of FCoV infection does prevent FIP.

## 2. Materials and Methods

### 2.1. Cats

For this retrospective study, we searched the database of one of the authors (DDA) for households of cats from which FCoV had been eradicated and contacted the cat guardians to find out what happened to their cats. The criteria for including a household were that all the cats had FCoV RT-PCR faecal tests both before and after the antiviral course and a follow-up period of at least 6 months. The first criterion was that all the FCoV-infected cats (initially identified by positive RT-PCR test results of faecal samples) had to have eliminated the virus, as demonstrated by negative faecal FCoV RT-PCR tests after treatment, and that the cats which had initially tested negative for FCoV RNA remained negative; in other words, the criterion was that the uninfected cats had not become infected by their FCoV-shedding housemates prior to the latter stopping shedding the virus.

Previous research showed that the probability of dying from FIP was greatest in the first 6 months after the first exposure to FCoV [18]. Consequently, the second criterion for inclusion was that only households with at least 6 months of follow-up were included so that adequate time was allowed for incubating FIP, if present, to emerge. Households with insufficient data or a follow-up period of less than 6 months were rejected on the basis that not enough time had elapsed for FIP to develop because of the long incubation period, especially for non-effusive FIP; our concern was that there may have been cats whose gut had been cleared of FCoV, but in whom the virus could still be present systemically. Twenty-seven households containing 147 cats that had sufficient follow-up were included, and 118 (80%) of the cats were from 17 (63%) households which had at least 18 months of follow-up. In a previous field study, 73% of FIP deaths occurred within 18 months of the first known FCoV exposure [18].

### 2.2. FCoV RT-qPCR

A FCoV reverse transcriptase quantitative polymerase chain reaction (RT-qPCR) was performed at the University of Glasgow Veterinary Diagnostic Laboratory, Scotland, as previously described, with the addition of a control for faecal PCR inhibitors [19,21]. Faecal samples from Household 24 were tested at the Veterinarmedizinisches Labor, University of Zurich, Switzerland, as previously described [22], and faecal samples from Household 27 were tested at the Zoologix Laboratory, CA, USA.

### 2.3. FCoV Antibody Titre

The majority of FCoV antibody titres were measured by immunofluorescent antibody test at the University of Glasgow Veterinary Diagnostic Services (VDS) Laboratory, Scotland, as previously described [10]; an antibody titre of zero meant there was no signal at the initial dilution of sera of 1:10. Doubling dilutions of the sample were stopped at 1:1280. The initial FCoV antibody tests of cats in Households 7 and 19 were performed at IDEXX Laboratories, Wetherby, England, where the upper cut-off was a dilution of the sample of 1:10,240, and subsequent tests on those cats were performed at VDS.

### 2.4. Determination of the Risk of FIP Having Ceased

The gold standard of proof that FCoV was eliminated from a cat’s body entirely was the reduction in his or her FCoV antibody titre to zero. A significant reduction in antibody titre was also taken to indicate that there was no longer a viral antigen present to stimulate the immune response; in other words, it was taken to indicate that the cat was no longer infected with FCoV and that the risk of FIP had passed (unless the cat would be re-infected in future, obviously).

It would not have been ethical to ask pet guardians to have their cats blood-tested to establish their FCoV antibody titre just for the purposes of this study, although we did request cat guardians to test leftover blood should the cat require a blood test for some unrelated reason. The second criterion for determining that a cat had truly recovered from FCoV infection was the survival of the FCoV-exposed cat without developing FIP for a minimum of 6 months (*n* = 133) and preferably over 18 months (*n* = 114) from the date of the virus being cleared. In our previous study of 820 cats in 73 households, most FIP cases occurred within 6 months of the cat’s first known exposure to FCoV; 73% (27 of 37) of FIP deaths occurred within 18 months, and by 36 months, the probability of the remaining cats not developing FIP was 95% [18]. In a study of 400 kittens from households with endemic FCoV infection, all FIP kitten deaths occurred before one year of age [10]. In a field-modified live intranasal FIP vaccine study of 609 apparently healthy young cats under 12 months of age, all 31 cats who died of FIP did so within 285 days (9.5 months) of the vaccine or placebo being administered [23]. The vaccine was initially called Primucell FIP and is now called Vanguard^®^ Feline FIP Intranasal (Zoetis, Parsipanny, NJ 07054, USA).

### 2.5. Statistical Analysis

Since this was a retrospective study, there was no placebo group; therefore, the data from a previous study of 73 households containing 820 cats were used as a retrospective control group [18]. The two cohorts were similar, being composed of a mix of ordinary pet households and a few cat breeders where in both groups the reason for the first testing for the presence of FCoV was the occurrence of FIP or FCoV enteritis.

The statistical significance of the results was calculated using an online Fisher’s exact test: https://www.socscistatistics.com/tests/fisher/default2.aspx (accessed 7 March 2023). A *p* value of <0.01 was considered highly significant and a *p* value of <0.05 was considered significant.

## 3. Results

### 3.1. Description of the Cohort of Cats, and Reasons for FCoV Elimination

FCoV was eradicated from 27 households containing a total of 147 cats (see Table 1). The majority of households were in the United Kingdom (UK), but Households 1, 9 and 27 were located in the United States of America and Household 24 was in the European Union. Household size ranged from one to 25 cats. At the time of treatment, their ages ranged from 8 weeks to 18 years, with 56 of 147 (38%) cats being under 2 years old and 36 (24%) of those being up to 12 months of age.

The most common reason for treating FCoV-infected non-FIP cats with antiviral drugs was to prevent the re-infection of a patient being treated for actual (*n* = 13) or suspected (*n* = 2) FIP (see Table 1). Five cat breeders wished to eradicate coronavirus following FIP deaths (Households 8, 10, 16, and 18), and a histopathologically confirmed outbreak of fatal FCoV enteritis in kittens (Household 27). In six households (3, 5, 9, 15, 19 and 27), cats were treated for FCoV-associated chronic enteropathy (CE). The remaining two households eradicated FCoV from their cats to prevent virus transmission to uninfected cats.

The retrospective control study of 73 UK households contained 251 (30.6%) adult cats and 569 (69.4%) kittens, of which 287 were FCoV antibody-negative and 282 had seroconverted. Monitoring for FCoV antibodies began because a cat presented with FIP (*n* = 33 households, containing 420 cats and kittens); 14 households with a total of 110 cats and kittens had re-homed a kitten which developed FIP, and the remaining 26 households containing 290 cats and kittens first tested for other reasons including the suspicion of FCoV enteritis (*n* = 3) or FIP (*n* = 4), with a routine pre-mating test (*n* = 7), and contact with a suspected FCoV excretor (*n* = 8) [18]. Household size in the study ranged from one to 42 cats; 45 households belonged to cat breeders [18]. At the time of that study, FCoV RT-PCR testing of faeces was not available to the authors; therefore, the presence of virus in the household was indirectly determined by FCoV antibody testing.

### 3.2. FCoV Elimination

Thirteen cats were treated for FIP. Faecal samples from 134 non-FIP cats were tested by RT-qPCR; FCoV RNA was detected in 109 cats without FIP and 25 cats tested negative.

Non-FIP cats positive for FCoV RNA in faecal samples were treated with 4–7 days of oral GS-441524 made by Bova Specials UK, Ltd., London, England (*n* = 3); Mutian^®^, Mutian Biotechnology Co., Ltd., Nantong, China (*n* = 98); or Panda, https://maxpawhealth.com (accessed 7 March 2023), USA (*n* = 8).

Previous studies showed that a dose of one pill of Mutian 100/kg given daily for a minimum of 4 days would stop FCoV shedding [19]. The Mutian 100 pill packaging claimed to contain 5 mg of active antiviral (subsequently shown to be GS-441524 [20]). However, an independent analysis of a Mutian 200 pill showed that it contained 18 mg of GS-441524, not 10 mg, as claimed (Nick Bova, personal communication), and an independent analysis of a Mutian 50 pill showed that it contained 7.2 mg of GS-441524, not 2.5 mg (Dominik Mirowski, personal communication). Consequently, for stopping FCoV shedding, Bova GS-441524 was administered at a dose of 10 mg/kg, not 5 mg/kg, as would have been expected had the Mutian 100 packaging been accurate. So far, as we know, no independent analysis of the contents of Panda pills has been conducted, but they seemed to stop virus shedding at the dose recommended on the seller’s website.

Faeces from all 134 cats plus the 13 cats with FIP being treated with oral antiviral drugs tested negative for FCoV RNA after antiviral therapy. Usually one course of treatment sufficed; if not, treatment was repeated until faeces tested negative for FCoV RNA by RT-qPCR. Twenty-five (17%) of the 147 cats did not shed FCoV in their faeces and therefore were not treated.

### 3.3. FCoV Antibody Titre Reduction

A reduction in FCoV antibody titre to zero was considered proof of recovery from FCoV infection, but very few FCoV antibody titres were available. A substantial reduction in the cats’ FCoV antibody titres occurred in 7 households (1, 2, 5, 6, 7, 8, and 19) containing 59 cats and is reported in Table 1’s comments section. FCoV antibody titres returned to zero in 9 cats, proving that they had eliminated FCoV, and there was a significant reduction in FCoV antibody titre in a further 9 cats. In an individual household, not all of the cats became seronegative within the relatively short time of our follow-up; the cats which had recovered from FIP often remained seropositive for over one year post-recovery even though they had stopped shedding the virus (we saw this in cats in Households 1 and 7 and have reported the long duration of the presence of FCoV antibodies in FIP-recovered cats previously [24]).

Two new kittens introduced into Household 23 remained seronegative one month after introduction, showing that they had not been exposed to FCoV; this was deemed an indirect proof of FCoV eradication from that household. These two kittens were not included in the survival statistics because they had not been exposed to FCoV infection. Kittens and cats born into, or introduced into, these FCoV-free households at later times were not counted in the statistical analysis because they failed to meet the inclusion criteria (i.e., proven exposure to FCoV infection plus a minimum of 6 months follow-up), but none developed FIP. Cat breeders were asked whether there had been any FIP amongst their kittens and, as far as they knew, none had occurred.

### 3.4. Survival of Cats after FCoV Elimination

In the present study, we included a follow-up for at least 6 months (a range of 6–43 months) for 27 households from the time of FCoV eradication from their households. In total, 11 of 147 cats died (Table 2). In the retrospective control group [18], the index FIP cases were not counted in the 820 cats; therefore, 13 FIP cases in this study were not included, leaving 134 cats.

In the retrospective control group, 37 of 820 (4.5%) cats died of FIP over the whole period of observation, and most FIP deaths occurred within the first 6 months from the first detection of FCoV/FIP in the household [18]. The Fisher exact test’s statistic value of the difference between 37 of 820 cats and 0 of 134 cats was 0.0062, which was highly significant at *p* < 0.01.

### 3.5. Survival of Cats for at Least 18 Months after FCoV Elimination

In the previous study, the probability of dying of FIP was greatest within the first 6 months; 2% of the retrospective cohort of 870 cats and kittens died of FIP within the first 6 months of FCoV first being detected in their household, and 27 of the 37 FIP deaths (73%) occurred within 18 months of FCoV first being detected in the household [18]. FIP did occur after 18 months, but by 36 months after the first FCoV diagnosis in a household, the probability of not developing FIP had increased to 95.2% [18]. Consequently, a subset of data from cats in the present study whose survival outcomes were known for at least 18 months after FCoV elimination was analysed separately. This subset of data is shown in the first 17 households in Table 1.

Overall, 8 cats from Households 1–17 died, but no cat died of FIP; the causes of death are shown in Table 2. Households 1–17 began with 118 cats, but 4 cats died of non-FIP-related causes before 18 months and were not counted; therefore, only 114 remained with at least 18 months of follow-up. Of the 114 remaining cats, 9 were successfully treated for FIP, but these were not counted in the Fisher exact test because in the first study cats with FIP died and so were not included [18]; this left 105 cats.

By 18 months in the retrospective control group, 27 FCoV-exposed cats had died of FIP and 572 were alive [18], whereas in the present study, 105 non-FIP cats remained at 18 months and zero cats had developed FIP. A difference between the 27 FIP deaths among the 597 cats followed previously for 18 months, and the occurrence of no deaths amongst the 105 cats alive at 18 months in the present study is considered significant; *p* = 0.023 (*p* < 0.05).

### 3.6. Comparison of Non-FIP Mortality between the Two Cohorts of Cats

In the present study, 11 of 147 (7.5%) cats died of non-FIP-related conditions, and in the retrospective control group, 87 of 820 (10.6%) cats died of causes other than FIP; although it appeared that fewer cats died in the present study, the difference was not statistically significant (*p* > 0.05).

### 3.7. Resolution of FCoV-Associated Chronic Enteritis in Eight Households

The cats in Households 3, 5, 9, 15, 19 and 27 were intentionally treated to eradicate FCoV in order to cure various chronic gastrointestinal (GI) signs that had been diagnosed as idiopathic inflammatory bowel disease (IBD) or chronic enteropathy (CE). GI signs were resolved in all cats, with the exception of chronic regurgitation in one cat in Household 9 that did not improve. Household 27 had lost kittens with histopathologically confirmed necrosuppurative enterocolitis with crypt and gland abscess lesions staining positive for FCoV (Veterinary Medical Diagnostic Laboratory, Colombia, Missouri, USA). No further kitten deaths occurred following FCoV eradication from this cattery.

The veterinary surgeons and/or guardians of the six households above knew or suspected that their cats’ CE signs were due to FCoV, but for two other cat guardians (Households 17 and 18), the recovery of their cats following the short course of GS-441524 was unexpected. Some guardians of FIP cats also reported a resolution of GI signs when FIP was treated (data not shown); enteritis in FIP cases has been previously reported [25]. Seven other CE recovered cats were from households which were excluded because follow-up lasted under 6 months or because there was no post-treatment FCoV RT-PCR test.

## 4. Discussion

While it was previously shown that a short course of an antiviral drug would stop FCoV shedding in the faeces, it was unknown whether this effect was temporary, or if the virus could linger elsewhere in the body allowing FIP to develop at a later stage [19]. Antibodies do not persist in the absence of an antigen; therefore, a return of the FCoV antibody titre to undetectable levels establishes that the virus has been eliminated from the body and that the risk of FIP is likely gone. Given that it would not have been ethical to request blood tests from pet cats for FCoV antibody testing, an alternative method to antibody testing for assessing FIP risk was to observe the cats for a period of time after which the risk of FIP would be deemed negligible. The present study combined FCoV antibody titre reduction (where available) and a long-term follow-up of FCoV-exposed cats to establish whether the risk of FIP had passed. We believe that we have demonstrated that the early elimination of FCoV infection—by which we mean treating the virus before the cat has developed FIP—is an effective method to prevent FIP.

The data from a previous study [18], performed before the advent of anti-coronavirus drugs, were used as a retrospective control group because it would also have been unethical to give placebos to pet cats to establish a prospective control group. The historical cohort was not perfectly matched to our cohort, but it was the largest available study on naturally infected cats in the field. Apart from antiviral use in only the present study, there were two major differences. First, the previous study could only indirectly infer the presence of the virus using FCoV antibodies, and the occurrence of FIP or FCoV enteritis as indicators of the presence of the virus, whereas in the present study we had the advantage of being able to accurately identify active FCoV infection by detecting viral RNA in the faeces; therefore, the present study more accurately identified households with endemic infection. In the historical cohort, over half of the cats in 24 of the 73 households became seronegative, indicating that coronavirus had died out in those households, which would have skewed the FIP mortality rate to be lower in the retrospective group since cats and kittens were included although they were no longer exposed to virus.

The proportion of cat breeders was higher, at 61.6% (45 of 73 households), in the retrospective control group [18] than in the present study, where it was 18.5% (5 of 27). The second and most important difference between the two studies was that kittens born into breeding households were included in the historical cohort of 820 cats, but over 129 kittens born after FCoV had been eradicated from their cattery were omitted from the present study because our major aim was to find out whether or not eliminating FCoV infection in FCoV-exposed cats would prevent FIP and the kittens in the present study had not been exposed to FCoV in their cattery of origin. Consequently, the proportion of kittens was higher in the control group than in the present study (69% vs. 24%). Kittens shed more of the virus than adult cats do; therefore, the environmental virus load would have been greater in the earlier study, leading to more FIP deaths. Around half the kittens (287 of 569) in the previous study were FCoV-antibody-negative, indicating that they had not been exposed to FCoV and none died of FIP; the FIP mortality rate in the FCoV-infected kittens was 7.8% (22 of 282 kittens died) [18].

Enrolment in the retrospective and present cohorts was similar—in both groups, cat guardians approached their veterinary surgeons either because their cat had FIP, because a kitten they had sold developed FIP, or because their cats or kittens suffered from chronic diarrhoea. There was a skewing of both groups towards pedigree cats compared with cats in the general population, because FCoV [8,26] and FIP in some studies [27,28], but not another [29], disproportionately affected pedigree cats. In summary, there were differences between the two groups that would affect the FIP incidence in either direction.

One concern could have been whether we had inadvertently chosen a population of cats in the present study which were already immune to FCoV infection, since the main risk of FIP follows the first exposure to FCoV infection; 2% of cats die of FIP within six months, and the risk of developing FIP falls to 4.8% at 36 months [18]. However, this was not the case, as 56 cats (38%) of the 147 cats were less than 2 years of age, and although cats of any age can develop FIP, around half of all FIP cases occur in cats under 2 years of age [10,27,28,29,30]. Furthermore, in two households, following the introduction of a pedigree kitten, it was the older incumbent cats, aged 3 and 7 years, rather than the kittens, who developed FIP.

A possible criticism of our study is that necropsies were not carried out on most of the cats who died. However, the guardians and veterinary surgeons of these cats had experience with FIP and would have been very alert to the possibility of it appearing and able to recognise it had it occurred. FIP was suspected in one cat in Household 7 when an in-contact cat of the FIP survivor presented with tenderness over the lumbar spine. For this reason, his blood was tested, and a significant reduction in his FCoV antibody titre provided reassurance that he did not have FIP; he recovered uneventfully with symptomatic treatment.

The incubation period for FIP, from infection to the appearance of clinical signs, may be a few weeks, but can be many months or, rarely, years [31]. In an experimental infection of laboratory cats who were euthanised 80 days after infection, FCoV was found in almost all internal organs; the cats had appeared to be healthy apart from transient pyrexia shortly after infection [32]. In a UK study, the virus was found in tissues in 22% of cats without FIP, compared with 95% of cats with FIP [33]. In a field study of the intranasal FIP vaccine, 31 of 609 cats under one year old developed FIP within 285 days of the placebo or vaccine being administered; viral RNA was detected in stored blood samples taken at the time of vaccination [23], therefore they were already systemically infected and were incubating FIP even though they seemed well enough to vaccinate. These studies showed that in cats who appear to be healthy, the virus can be present systemically, not just in the gastrointestinal tract; therefore, we expected that in our group of treated cats there might have been some that were incubating undetected FIP, for whom a short course of an antiviral would not be sufficient, since most cats with FIP require around 7–8 weeks [24], 12 weeks [34,35] or even—in relapsed cases—two courses [34,35,36] of treatment. However, FIP did not occur in cats whose intestinal FCoV had been eliminated, possibly because our cohort of cats was too small to detect such an event, although antivirals given orally do also act systemically.

FCoV can cross the blood–brain barrier, resulting in neurological signs; our biggest concern was that the amount of antiviral required to stop FCoV shedding is around half the dose required to eliminate FCoV from the brain [24]; therefore, we were especially watchful for neurological FIP development. FIP treatment relapses frequently present with neurological signs [36].

The main site of FCoV replication is the intestine [7]: therefore oral antivirals are more effective in eradicating the virus from the body than injectable drugs because oral antivirals go directly to the site of major virus replication. In FCoV infection, drug-resistant viruses are more likely to develop in cats with FIP being treated with injectable antivirals which do not adequately penetrate the gut. In our experience, injectable GS-441524 does not consistently eliminate FCoV from the gut; one cat successfully treated for FIP with GS-441524 injections for 12 weeks still shed the virus in his faeces for at least two years post-treatment [24]. It is from such carrier cats, treated with injectable instead of oral antivirals, that virus mutants resistant to GS-441524 are most likely to emerge. In a recent study of 26 cats who experienced FIP treatment relapses, 23 were treated using injectable rather than oral GS-441524 [36], but unfortunately, no control group of cats without relapses was presented to prove that the reason for the relapses was beginning treatment by injection. 

In many countries, GS-441524 is available directly to the public; there is a risk that if FCoV elimination or FIP treatment is not performed at a high enough dose, or not performed for a long enough duration, then virus resistance to the drug could ensue. It is essential that a post-treatment faecal RT-PCR test be performed to ensure that virus elimination has been effected.

Most households were small (fewer than 10 cats), and the eradication of FCoV from the household was straightforward. However, virus elimination from larger households, and especially households with kittens, was more challenging, because FCoV is highly contagious, and if even one cat did not eliminate infection, he or she might have rapidly re-infected the other cats with whom he shared litter trays [19]. Indirect virus transmission via cat litter dust fomites is also a risk. One cat breeder with a very large cattery who eradicated FCoV from her cats, unfortunately, re-introduced the virus by bringing in a large number of FCoV-infected pedigree kittens. As expected, in that household, the re-infected cats began to shed the virus again and further FIP cases occurred (data not shown). This household is a reminder that once FCoV is eradicated from a household, strict precautions must be put in place to prevent its re-introduction. Within the households that failed to meet the inclusion criteria, this household was the only one to experience FIP in addition to their original case. The breeder reported that the re-introduction of FIP into her cattery was especially serious in kittens, presumably because they had no maternally derived antibody protection; there is a precedent for this observation in an experimental study where 20 FCoV-naïve kittens were introduced into two households with endemic FCoV, resulting in a 90% mortality of the kittens within 100 days [37]. Four households also introduced infected kittens or cats post-FCoV eradication, but the new animals were quarantined, faecal- or antibody-tested and treated, if necessary, before being allowed to share litter trays with uninfected cats, and no virus transmission occurred.

Another possible criticism of this work is that eliminating the virus from the body artificially may interfere with the development of natural immunity. That may be the case, but should be weighed against the fact that FIP was prevented and that re-exposure to FCoV infection can be avoided if virus is eradicated from the entire household of cats.

The elimination of FCoV cured some cats in the study that were suffering from chronic gastrointestinal conditions (chronic is defined as occurring for a duration of over three weeks [12,13], and in our cohort, some cases had suffered for years). The presence of FCoV in an infectious disease diarrhoea profile tends to be overlooked because the virus is highly prevalent in purebred cats [8,26] and multi-cat environments such as cat shelters [38,39,40]. Just as any good gastroenterologist would eliminate Giardia, *Tritrichomonas foetus*, or any other known diarrhoea-causing pathogens before conducting a more invasive procedure such as endoscopy or biopsy, the recovery of several cats in our small cohort shows that a response to an oral (not systemic) anti-coronavirus drug should now be included amongst the therapeutic trials performed in suspected FCoV-infected feline IBD cases where the stools remain loose despite the elimination of other diarrhoea-causing pathogens, and where there is a lack of response to dietary changes and probiotics.

## 5. Conclusions

In conclusion, cats that were infected with FCoV (but did not have FIP) were treated with a short course of an antiviral drug. The treatment was safe, cleared the virus from the gastrointestinal tract, and prevented the future development of FIP. The role of FCoV in cats suffering from chronic gastrointestinal disease requires further investigation.

## Figures and Tables

**Table 1 viruses-15-00818-t001:** Description of 147 cats in 27 households from which FCoV was eradicated.

Household	No. of Cats in Household	No. of FIP Cats Treated	No. Non-FIP Cats Cleared of FCoV after 4–7 d Antiviral	Duration of Follow-Up Post-Antiviral in Months	No. of Cats Developed FIP	Comments
**1**	4	1	2	43	0	Itraconazole (Itrafungol, Virbac, Fort Worth, TX 76161, USA) treatment suppressed viral load only while cats were taking it, but failed to eliminate the infection, so cats were subsequently treated with Mutian pills. Faeces of all four cats remained negative for FCoV RNA 11 mo post-FCoV-eradication. FCoV antibody titres reduced significantly in two cats (1280 to 80 and >1280 to 320) within 4 mo. FCoV antibody titres reduced insignificantly (>1280 to 1280) in a third cat but remained high in the FIP recovered cat for over 2 y.
**2**	17	0	13	42	0	13 of 17 cats were treated following the death of one cat which the histopathology report described as being highly suspected to have FIP. A second cat suspected of suffering from FIP in fact had lymphoma (see Table 2). FCoV antibody titre of four cats reduced to zero and that of a fifth cat reduced from 640 to 20. One cat tested at 42 mo was seronegative.
**3**	7	0	3	26	0	Chronic diarrhoea in one young cat was resolved following treatment.
**4**	3	1	2	26	0	
**5**	4	0	2	27	0	One chronic FCoV-diarrhoea kitten was introduced from a rescue shelter; all four cats got diarrhoea, although all had FCoV antibodies the faeces of only one other cat tested FCoV RNA positive. Reduction in FCoV antibody titre from >1280 to 80 was observed within 13 mo in the chronic-diarrhoea-recovered cat. The FCoV antibody titre was 40 at 5 mo post-treatment, then 0 at 10 mo post-treatment in the other FCoV-shedding cat, and 0 and 40 at 5 mo post-FCoV-eradication in the two non-shedding cats.
**6**	2	1	1	24	0	FCoV antibody titre of non-FIP cat reduced from 640 to 80 after 14 mo.
**7**	5	1	3	30	0	FCoV antibody testing was performed 7 mo after FCoV had been eradicated because a second cat presented with clinical signs for which FIP was a differential diagnosis. FIP was eliminated as a diagnosis because his FCoV antibody titre had reduced from >10,240 to 40. The FCoV antibody titre reduced from >10,240 to 80 and 160 in two other cats but remained at 640 for 12 mo in the fourth cat, although no faecal shedding was detected, reducing to 160 at 16 mo. FCoV antibody titre reduced from >10,240 to 640 in 9 mo in the FIP-recovered cat.
**8**	25	0	20	43	0	Active breeding cattery with cats kept in small groups. One adult cat became FCoV seronegative (others were not tested). Four 11 wk old kittens tested for FCoV antibodies 6 mo post-FCoV-eradication: all were seronegative. A total of 102 kittens were born into this cattery since FCoV was eliminated: none developed FIP.
**9**	3	0	2	36	0	Both FCoV-infected cats had a 3 y history of chronic intermittent diarrhoea, and positive FCoV RT-PCR tests over a 3 y period; all three cats had FCoV antibodies. Infectious causes of diarrhoea other than FCoV were ruled out 2 y previously (RealPCR diarrhoea panel, IDEXX laboratory, Sacramento, California, USA), except for a discordant PCR positive/antigen negative test for Giardia 3 y previously in one cat. Treatment was attempted with metronidazole; fenbendazole; tylosin; probiotics (Proviable^®^, Nutramax, Lancaster, SC 29720, USA then Visbiome^®^, ExeGI Pharma, LLC, Rockville, MD 20850, USA); Food Sensitivities’ venison/green pea food (Hill’s d/d, USA). Itraconazole treatment reduced coronaviral load but failed to eradicate the infection. Subsequent treatment with GS-441524 pills resolved the chronic diarrhoea, although one cat continued to regurgitate regularly.
**10**	20	0	20	34	0	16 adults and four 4 mo old kittens were cleared of FCoV following the death of two cats from FIP. Two new FCoV-infected kittens introduced into the household were quarantined and cleared of coronavirus but are not counted in this table or statistics because follow-up was less than 6 mo at time of writing.
**11**	2	1	1	26	0	The non-FIP in-contact cat was alive and well at 26 mo, but the FIP-recovered cat developed cancer (see Table 2).
**12**	3	1	0	23	0	Only the FIP patient (biopsy and ascites FCoV RT-PCR-confirmed) required antiviral treatment; the faeces of the two in-contact cats were negative.
**13**	2	1	1	21	0	
**14**	2	1	1	19	0	
**15**	3	0	2	19	0	The index case was an 11.4 y old female Maine Coon cat who was presented to her new veterinary surgeon with a history of chronic diarrhoea believed to be due to chronic exocrine pancreatic insufficiency requiring daily pancreatic enzyme capsules (Lypex, VetPlus, Lytham, Lancashire, UK) sprinkled on food. However, feline pancreatic lipase was normal (2.3 μg/L), and cobalamin was normal but folate was below normal at 4.3 μg/L. Her faecal FCoV viral load was moderate (C*_T_* was 28; FCoV antibody titre was moderate at 160; alpha-1 acid glycoprotein was raised at 1919 μg/mL). Chronic FCoV enteropathy was diagnosed, and she recovered completely following elimination of the coronavirus.
**16**	13	0	13	18	0	Breeding household: 10 litters born prior to FCoV eradication contained 29 kittens, of which two were stillborn and three faded. FIP was suspected but not histopathologically confirmed in two sold kittens and two kept kittens. A total of 23 living kittens (and one stillborn) in 6 litters were born since FCoV was eradicated; all 23 are alive and well.
**17**	3	1	1	18	0	The non-FIP FCoV-infected cat was described by her guardian as having had chronic gastrointestinal disease for her whole life (she was 8 y old) which ceased following antiviral treatment.
**18**	3	0	2	16	0	A Ragdoll cat breeder treated her cats to clear FCoV and was surprised that one cat with chronic diarrhoea recovered following treatment with GS-441524. All three cats remained well 16 mo after FCoV eradication from the household and one had a litter of three kittens.
**19**	2	0	2	15	0	A history of chronic intermittent diarrhoea which was resolved after treatment with GS-441524. The FCoV antibody titre of both cats reduced to 0 from 2560 and >10,240 within 3 and 7 mo respectively.
**20**	2	1	1	10	0	
**21**	4	1	3	15	0	
**22**	2	1	1	15	0	
**23**	4	0	2	13	0	Two new kittens were introduced 12 mo after FCoV eradication; both tested FCoV-antibody-negative after being in the household for over one month, showing that they had not been exposed to virus.
**24**	2	1	1	11	0	
**25**	1	0	1	8	0	Uveitis was erroneously suspected to be due to FIP.
**26**	1	0	1	9	0	The treated cat was in contact with a FIP cat who died; his guardian wishes to purchase a new kitten.
**27**	8	0	8	6	0	Prior to clearing FCoV, an 8 wk old kitten died of histopathologically confirmed necrosuppurative enterocolitis with crypt and gland abscesses; the lesions stained positive for FCoV. Five litters were born post-FCoV-eradication; this breeder spot checks kittens for FCoV shedding, and they have been negative. Two new FCoV-infected kittens were introduced into the household and cleared of infection since the original FCoV eradication (they are not counted in the statistics because their follow-up lasted under 6 mo).
	147	13	109		0	Total

No. = number, mo = months, wk = week(s), y = year(s).

**Table 2 viruses-15-00818-t002:** Causes of death in 11 cats who died after FCoV eradication from their households.

House-Hold Ref No.	No. of Cats Died	Interval from FCoV Eradication from Household to Death in Months	Comments
2	2	10 mo: tumours 29 mo: tumours	Two cats died in the 3 y follow-up; one died at 10 mo (13 y 11 mo old) and one at 29 mo (6 y 10 mo old) after the household was cleared of FCoV; both cats had tumours diagnosed prior to FCoV eradication. One died following a kidney infection related to being paraplegic and doubly incontinent for 5 y and the other developed autoimmune haemolytic anaemia following chemotherapy for lymphoma.
8	1	24 mo: sudden death.	One 10 y old cat died suddenly, having been fine in the morning and eaten his breakfast. No post-mortem was conducted.
10	1	30 mo: HCM	One British shorthair death occurred due to hypertrophic cardiomyopathy (HCM) 30 mo after the household was cleared of FCoV: his son also died of HCM but was not included in statistics since he was born after FCoV was eradicated from the household.
11	1	14 mo: throat tumours	The cat treated for FIP developed throat tumours and died under GA 14 mo after FIP treatment.
15	1	21 mo: acute kidney failure	One sudden death occurred at 21 mo of a 4 y old Maine Coon with acute kidney failure.
16	1	14 mo: sudden death post-kittening	Breeding household: the single death since FCoV was eradicated was a sudden death 4 wk post-kittening; the cat went into seizure and was euthanised at an emergency vet centre. No abnormalities were detected in the post-mortem.
17	1	10 mo: cancer	The cat who died of cancer was not the cat who was treated for FIP.
23	2	9 mo: old age 10 mo: road accident	At 9 mo post-FCoV-eradication, a very old cat was euthanised, and at 10 mo, a young cat was killed on the road.
27	1	3 mo: sudden death	A 9 mo old Ragdoll cat died suddenly 3 mo after FCoV was eradicated; she had a history of a heart murmur. Her necropsy reported a normal body condition score (5–6/9) and a weight of 8.1 pounds; kidney disease was suspected because her kidneys were small but distended; heart size appeared normal. No histopathology was performed but FIP lesions were not seen in a gross post-mortem.
**9**	**11**	**Total**	

No. = number, mo = months, wk = week(s), and y = year(s).

## Data Availability

Not applicable: all relevant data is included in the paper.

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
