# Peer review of "Stopping Feline Coronavirus Shedding Prevented Feline Infectious Peritonitis"

_viruses, 2023, doi:10.3390/v15040818_

Round 1

Reviewer 1 Report

In the present study the Authors suppose that early treatment of FCoV-infected cats with oral antivirals prevented FIP. The hyphotesis seems to be interesting but it based on misconceptions of virology and immunology. As weel as the methods to evaluate their thesis are arbitrary.certainly, antiviral therapy has its concrete merits in eradicating a virus but, in the present study the inclusion criteria for including the cats were that all the 80 FCoV-infected cats had negative faecal FCoV RT-qPCR tests after treatment and that only households with at least 6 months follow-up. The Authors does not remember that FCoV infection is uquitous and reinfection are as frequent as common. The study should be carried out on FCoV infected cats but kept in isolation.

The Authors report that reduction of FCoV antibody titre to zero was considered proof of recovery from FCoV infection and that FCoV antibody titres returned to zero in 9 cats, proving that they have eliminated FCoV. Cat with natural infection developed an immune response with memory cells and the serology does not allow to conclude recovery from infection.

The study can only demonstrate that antiviral therapy can temporary stop FCoV shedding, but it cannot be considered a prevention for FIP development

Author Response

DDA et al: first we would like to thank you very much for taking the time to read and rapidly review our paper.

Reviewer 1

In the present study the Authors suppose that early treatment of FCoV-infected cats with oral antivirals prevented FIP. The hyphotesis seems to be interesting but it based on misconceptions of virology and immunology.  As weel as the methods to evaluate their thesis are arbitrary.certainly, antiviral therapy has its concrete merits in eradicating a virus but, in the present study the inclusion criteria for including the cats were that all the 80 FCoV-infected cats had negative faecal FCoV RT-qPCR tests after treatment and that only households with at least 6 months follow-up. The Authors does not remember that FCoV infection is uquitous and reinfection are as frequent as common.

DDA et al replied: actually 118 cats (87 positive for FCoV) had a minimum of 18 months follow-up, and only 29 cats (22 infected) had a minimum of 6 months, but less than 18 months of follow up.  We wanted to make sure that we gave as long as possible post-infection for the development of FIP, if it was incubating in any of the cats. In addition to the criterion of post-treatment negative RT-qPCR you correctly mention, the criteria included proven positive FCoV RT-PCR testing prior to treatment, as well as negative testing after treatment, which we have now made clearer in the paper.

We do indeed remember that FCoV infection is ubiquitous, we are very much aware of that having studied field cases for over three decades, but re-infection of a cat can only occur if they share a litter tray with an infected cat, or are exposed to faecal fomites.  In the field situation we did not find that re-infection was a big problem, because the 27 household owners involved were very knowledgeable and were able to prevent re-infection of their cats, even in those households where new infected cats were introduced (with the exception of one very large household mentioned in Discussion).

Reviewer 1: The study should be carried out on FCoV infected cats but kept in isolation.

DDA et al reply: I would like to know the rationale for this suggestion because I’m sorry I haven’t been able to work out the reason?  Unfortunately, in a retrospective observational field study it is not possible to conduct such a study, or at least we are not able to, with our limited resources.   The aim of our work is to obtain information applicable to the real world; therefore the observation of naturally infected cats in normal pet households gives data that is most useful to veterinary surgeons in practice which is our target audience.

Reviewer 1: The Authors report that reduction of FCoV antibody titre to zero was considered proof of recovery from FCoV infection and that FCoV antibody titres returned to zero in 9 cats, proving that they have eliminated FCoV. Cat with natural infection developed an immune response with memory cells and the serology does not allow to conclude recovery from infection.

DDA et al reply: we know that FCoV carrier cats have high FCoV antibody titres from constant stimulation of the immune system, and having followed literally thousands of naturally infected cats, we have never seen a cat whose antibody titre returned to zero (i.e. less than a 1in 10 dilution) spontaneously begin shedding virus again with no possible external source of re-infection.  Unless a cat is spectacularly immunosuppressed for some reason (e.g. chemotherapy), cessation of an antibody response equates to the cat having eliminated viral antigen from his or her body: yes, memory cells persist, but antibodies are only produced again if the cat becomes exposed to the antigen again. 

Reviewer 1: The study can only demonstrate that antiviral therapy can temporary stop FCoV shedding, but it cannot be considered a prevention for FIP development

DDA et al reply: the evidence we present shows with high statistical significance that early FCoV treatment did prevent FIP: we would have expected to see 8 or 9 FIP cases within the cohort of 134 non-FIP cats, and another 12 amongst the kittens born in the breeding catteries, with the caveat that FIP can still occur should re-infection occur.  In other words, the protocol we describe is not the same as a vaccine, but is more like malaria prevention, or is like stopping SARS-CoV2 infection progressing to COVID by early treatment.

Reviewer 2 Report

This study reports the incidence of FIP related disease in 27 households/147 cats following antiviral treatment with 4 to 7 days of oral 141 GS-441524 and supposed elimination of gastrointestinal FCoV infection. It then compares this incidence to a historical control group of 820 cats to demonstrate improved survival, presumably due to the antiviral treatment.

I would like to commend the authors on their efforts. I believe more field studies like this need to be performed, as they are the best litmus test for what is actually happening in feline medicine in the real world. Field studies are notoriously messy and challenging to perform, and this can make writing and reading these studies also challenging.

I think there is merit in the study these authors have reported here. I’m sorry to say I found the manuscript overall difficult to read. In some sections I found it bereft of detail, and in other sections I found the writing style too simplistic and dumbed down. Other times I just wanted to reach into the manuscript and plead with the authors to be explicit about what they were wanting to say. Of particular concern for me was I couldn’t understand precisely what testing had been performed, nor the results obtained, and these need to be obvious for even dummies like me to follow.

- Section 2.1 – Where were the households located? A map would be very helpful. UK and Europe and America? What effect might this possibly have on results, given possible geographical FCoV strain variations?

- ‘The criteria for including a household were first that all the FCoV-infected cats had negative faecal FCoV RT-qPCR tests after treatment…’ BUT ‘Re-infection and virus shedding at the limits of detection, and/or inhibition of polymerase chain reaction (PCR) tests by faecal inhibitors [6] can give an erroneous impression of intermittent virus shedding.’ So the authors need to give more information about how many times cats were tested in order to demonstrate elimination of FCoV infection

- Section 2.2 – Do references 16 and 18 also relate to the RT-PCR testing performed at the Zoologix Laboratory, California, USA, and at the Veterinarmedizinisches Labor, University of Zurich, Switzerland? It would be helpful to list here a limit of detection for the assay at each institution

- Section 2.3 – Does reference 19 also relate to FCoV antibody testing performed at the University of Glasgow Veterinary Diagnostic Laboratory, Scotland? It would be useful to record here how antibodies are measured (is it serial dilutions starting at 1:2?) and what positive/negative results were recorded as

- L130 – Remove full stop after Table 1

RESULTS

- ‘FCoV was eradicated from 27 households containing a total of 147 cats’ – I don’t understand what testing was performed to demonstrate this. The testing performed was impossible to follow. How many of the 147 cats had RT-PCR testing performed? What tissue was RT-PCR testing performed on? I think only faecal samples but please make explicitly clear. E.g. Faecal samples from all 147 cats tested RT-PCR negative, on xxx occasions, as per the study inclusion criteria.

- ‘The gold standard of proof that FCoV was eliminated from a cat’s body entirely was reduction of his or her FCoV antibody titre to zero...‘ BUT ‘It would not have been ethical to ask pet guardians to have their cats blood tested for the purposes of establishing their FCoV antibody titre…’ So how many of the 147 cats had FCoV antibody testing performed? And why? Antibody testing was not one of the 2 criteria listed in Section 2.1 for study recruitment. I don’t understand how it ties in. Weren’t all cats to be included RT-PCR negative/antibody negative? (or at least reducing antibody levels). What if a cat was RT-PCR negative/serology positive, what happened?

- Section 3.1. and elsewhere. Is FCoV ‘elimination’ too strong? What about FCoV localisation elsewhere from the GIT without faecal shedding? My quick reading of other papers is that this can occur?  

- I couldn’t really follow Table 1 – there was a huge amount of information. Shouldn’t it go later in Results? And is it imperative, or could it be a Supplementary Table to improve the article flow?

- Sections 3.3-3.5. I would like to be more convinced that the control group used (reference 15) was a suitable control group. For example follow up in the present study was a period of at least 6 months, where as reference 15 was ‘conducted over a period of 6 years’. What about a statistical comparison of ages, sexes, breed etc. between the 2 groups?

- ABSTRACT ‘Cats from 8 households recovered from chronic FCoV enteropathy….’ RESULTS 3.6. ‘Resolution of FCoV-associated chronic enteritis in 8 households’ CONCLUSION ‘The treatment also cured a number of cats that were suffering from chronic gastrointestinal disease.’ I would like a little more evidence for this bold statement about ‘curing’ a chronic disease, to make me believe it. What other work up was performed in these 8 cats, prior to antiviral treatment? And what sort of follow up for each cat?

DISCUSSION

- Any other variables between the current cohort and the historical control group, apart from antiviral treatment, that might have been a factor here? Eg. Different husbandry? Quarantine of shedding animals?

- L248-250 These were not closed colonies, so how was the risk if re-infection accounted for? Lines like this in the Discussion left me doubting the results and wanting more information on how much testing was performed in order to be certain of FCoV eradication from a household. ‘As expected re-infected cats began to shed virus again and further FIP cases occurred (data not shown).’ Perhaps this data should be shown!

- L238-240 ‘We expected that in our group of treated cats there might be some that were incubating undetected FIP, for whom a short course of antiviral would not be sufficient, since most cats require around 7-8 weeks [20] or even 12 weeks of treatment [21].’ I think this finding needs to be expanded upon further. Presumably the difference is due to eradicating FCoV from the GIT of healthy cats, versus eradicating systemic FIP from sick cats. But I think for the reader’s benefit this difference needs to be emphasised

- What is your evidence of ‘early’ elimination of FCoV infection, in a field study without knowledge of day of challenge? ABSTRACT L28, DISCUSSION L219

- Some of the writing in this paper I found a little casual and not as sophisticated as I would have expected – e.g.  ‘Making antibodies costs the body energy, and the globulins themselves thicken the blood, therefore the body stops making antibodies as soon as it deems it safe to do so, but keeps memory T cells with the blueprint so it can produce antibodies again if need be.’ Sounds like it has been written for high school biology students to understand rather than intelligent veterinarians and virologists

- L276-277 ‘…response to an oral (not systemic) anti-coronavirus drug should now be included amongst the therapeutic trials performed in suspect feline IBD cases.’ I think this statement needs more qualification. When exactly are the authors advocating for a trial of FCoV antiviral medication? What testing should be performed prior to a treatment trial? What clinical signs (type, duration etc.) would prompt these authors to commence treatment?

What explicitly are these authors suggesting – should all kittens be treated prophylactically with antiviral medication? All kittens/cats rehomed from multi-cat households/shelters? What is the practical outcome of this study?

REFERENCES

- 22 references (of which 6 references are from the primary author) is underwhelming for an area of veterinary science in which there are many, many publications

Author Response

Reviewer 2: This study reports the incidence of FIP related disease in 27 households/147 cats following antiviral treatment with 4 to 7 days of oral 141 GS-441524 and supposed elimination of gastrointestinal FCoV infection. It then compares this incidence to a historical control group of 820 cats to demonstrate improved survival, presumably due to the antiviral treatment.  I would like to commend the authors on their efforts. I believe more field studies like this need to be performed, as they are the best litmus test for what is actually happening in feline medicine in the real world. Field studies are notoriously messy and challenging to perform, and this can make writing and reading these studies also challenging.

DDA et al reply: we are truly grateful for the amount of time and effort you put into this review and for your generous appreciation of our efforts.  We feel encouraged by your kind comments.  

Reviewer 2: I think there is merit in the study these authors have reported here. I’m sorry to say I found the manuscript overall difficult to read. In some sections I found it bereft of detail, and in other sections I found the writing style too simplistic and dumbed down. Other times I just wanted to reach into the manuscript and plead with the authors to be explicit about what they were wanting to say. Of particular concern for me was I couldn’t understand precisely what testing had been performed, nor the results obtained, and these need to be obvious for even dummies like me to follow.

DDA et al reply: you seem to have the opposite of the Dunning-Kruger effect because a dummy is clearly the last thing you are! : )   Which makes it all the more of concern that you found the manuscript difficult to read: thank you that you took the time and trouble to single out those parts which were least easy to understand.  We have taken on board all you said and have expanded much of the paper and we hope it is now easier to follow. 

Reviewer 2: - Section 2.1 – Where were the households located? A map would be very helpful. UK and Europe and America? What effect might this possibly have on results, given possible geographical FCoV strain variations?

Reviewer 2: DDA et al reply: This is a good question: presumably you’re thinking about the higher prevalence of type II FCoV in eastern countries and the higher incidence of FIP in type II infection?  We have put the following sentence into the methods, but with only 27 households, we didn’t feel a map was justified:

“The majority of households were in the United Kingdom, but Households 1, 9 and 27 were located in the United States of America and Household 24 in the European Union.”

We believe that since none of the households was in the east that geographical location is unlikely to affect our results.  I remember Dr Harry Vennema at a conference saying that coronaviruses from different locations were more similar to viruses from that location, rather than that viruses which had caused FIP were in any kind of cluster. We are very much aware that access to legal nucleoside antiviral drugs is different geographically and we are cheering Nick Bova on his way as he negotiates with various governments to have his GS-441524 pills made widely legally available to the veterinary profession.

Reviewer 2: - ‘The criteria for including a household were first that all the FCoV-infected cats had negative faecal FCoV RT-qPCR tests after treatment…’ BUT ‘Re-infection and virus shedding at the limits of detection, and/or inhibition of polymerase chain reaction (PCR) tests by faecal inhibitors [6] can give an erroneous impression of intermittent virus shedding.’ So the authors need to give more information about how many times cats were tested in order to demonstrate elimination of FCoV infection

DDA et al reply: the purpose of the sentence that you quoted here was to show how our view has evolved since my 2001 publication: experience of monitoring thousands of virus shedding cats since that time, using updated, more sensitive, RT-qPCR testing, has forced me (DDA) to admit that I was wrong about intermittent virus shedding.  Thanks to your remark, I have expanded that sentence to make it clearer.

“We previously believed that intermittent coronavirus shedding could occur [4] but now our view is that this belief was erroneous and that the appearance of intermittent virus shedding was due to re-infection (usually by infected cats in the same household); or virus shedding at the limits of detection of the assay concerned (a nested RT-PCR was used during our previous publication [4]); and/or inhibition of polymerase chain reaction (PCR) tests by faecal inhibitors [6].“

Where available, information about either repeat FCoV RT-PCR or FCoV antibody testing is already given in the comments section of Table 1.  As you said above, field observational studies are “notoriously messy” and we felt we had already imposed upon the good will of the household owners in having them reply to our enquiries on what had happened to their cats: it would not have been reasonable to expect them to re-submit faecal samples as well. 

Reviewer 2: - Section 2.2 – Do references 16 and 18 also relate to the RT-PCR testing performed at the Zoologix Laboratory, California, USA, and at the Veterinarmedizinisches Labor, University of Zurich, Switzerland? It would be helpful to list here a limit of detection for the assay at each institution

DDA et al reply: Section 2.2 has been amended in the hope that it is now clearer.  I regret that a limit of detection for each assay is not available from the three laboratories, but  Glasgow and Zurich Vet School laboratories tests are sensitive and are generally held to be gold standard laboratories in the UK and Switzerland.

Reviewer 2: - Section 2.3 – Does reference 19 also relate to FCoV antibody testing performed at the University of Glasgow Veterinary Diagnostic Laboratory, Scotland? It would be useful to record here how antibodies are measured (is it serial dilutions starting at 1:2?) and what positive/negative results were recorded as

DDA et al reply: Yes reference 19 is a description of the Glasgow University VDS FCoV antibody test and the description has been expanded upon to the extent that is relevant for the present publication.  The first antibody tests on cats in Household 7 were tested at IDEXX laboratories but unfortunately we do not know their techniques: subsequent tests from that household were performed at VDS.

“The majority of FCoV antibody titres were measured by immunofluorescent antibody test at the University of Glasgow Veterinary Diagnostic Services (VDS) Laboratory, Scotland as previously described [19]: an antibody titre of zero meant there was no signal at the initial dilution of sera of 1:10. Doubling dilutions of the sample were stopped at 1:1280.  The initial FCoV antibody tests of cats in Household 7 were performed at IDEXX Laboratories, Wetherby, England, where the upper cut-off is a dilution of the sample of 1:10,240 and subsequent tests on those cats were performed at VDS.”

Reviewer 2: - L130 – Remove full stop after Table 1

DDA et al reply: done.

Reviewer 2: RESULTS

- ‘FCoV was eradicated from 27 households containing a total of 147 cats’ – I don’t understand what testing was performed to demonstrate this. The testing performed was impossible to follow. How many of the 147 cats had RT-PCR testing performed? What tissue was RT-PCR testing performed on? I think only faecal samples but please make explicitly clear. E.g. Faecal samples from all 147 cats tested RT-PCR negative, on xxx occasions, as per the study inclusion criteria.

DDA et al reply: I’m very sorry that this was unclear.  Faeces from all of the 147 cats were tested before and after the antiviral course.

We have added to this sentence in 2.1 to make this more obvious:

“The criteria for including a household were that all the cats had FCoV RT-PCR faecal tests both before and after the antiviral course and a follow-up period of at least 6 months.  The first criterion was that all the FCoV-infected cats (initially identified by positive RT-qPCR test results on faecal samples) had to have eliminated the virus, as demonstrated by negative faecal FCoV RT-qPCR tests after treatment, and that the cats which had initially tested negative for FCoV RNA remained negative: in other words, that the uninfected cats had not become infected by their FCoV shedding housemates prior to the latter stopping shedding virus..” 

In addition, Section 3.1 has now been divided into two sections and hopefully what was done is now clearer:

“3.2 FCoV elimination

Thirteen cats were treated for FIP.  Faecal samples from 134 non-FIP cats were tested by RT-qPCR: FCoV RNA was detected in 109 cats without FIP and 25 cats tested negative. 

Non-FIP cats positive for FCoV RNA in faecal samples were treated with 4 to 7 days of oral GS-441524 made by Bova Specials UK, Ltd, London, England (n = 3);  Mutian®, Mutian Biotechnology Co., Ltd., Nantong, China (n = 98); or Panda, https://maxpawhealth.com, USA (n = 8). 

Previous studies showed that a dose of one pill of Mutian 100/kg given daily for a minimum of 4 days would stop FCoV shedding [19].  The Mutian 100 pill packaging claimed to contain 5mg of active antiviral (subsequently shown to be GS-441524 [20]).  However, independent analysis of a Mutian 200 pill showed that it contained 18mg of GS-441524, not 10mg as claimed (Nick Bova, personal communication), and a Mutian 50 pill contained 7.2mg, not 2.5mg (Dominik Mirowski, personal communication).  Consequently, for stopping FCoV shedding Bova GS-441524 was administered at a dose of 10mg/kg, not 5mg/kg as would have been expected had the Mutian 100 packaging been accurate.  So far as we know, no independent analysis of the contents of Panda pills has been conducted, but they seemed to stop virus shedding at the dose recommended on the seller’s website.

Faeces from all 134 cats plus the 13 cats with FIP being treated with oral antiviral drugs tested negative for FCoV RNA after antiviral therapy. Usually one course of treatment sufficed; if not, treatment was repeated until faeces tested negative for FCoV RNA by RT-qPCR.  Twenty-five (17%) of the 147 cats did not shed FCoV in their faeces and therefore were not treated.”

Reviewer 2: - ‘The gold standard of proof that FCoV was eliminated from a cat’s body entirely was reduction of his or her FCoV antibody titre to zero...‘ BUT ‘It would not have been ethical to ask pet guardians to have their cats blood tested for the purposes of establishing their FCoV antibody titre…’ So how many of the 147 cats had FCoV antibody testing performed? And why? Antibody testing was not one of the 2 criteria listed in Section 2.1 for study recruitment. I don’t understand how it ties in.

DDA et al reply: FCoV antibody testing ties into our study because demonstration of loss of antibodies is possibly the gold standard for recovery from FCoV infection: it shows that there is absolutely no antigen left anywhere in the body stimulating an immune response.  If all the cats in the study had become FCoV seronegative one wouldn’t have to follow the cats for years to find out if they were going to develop FIP or not: one would simply know that FIP was no longer a concern.  However, as we say, it would not have been ethical to ask people to put their cats through the invasive procedure of a blood sample, therefore we could only take note if they were having a blood test done under advisement of their own veterinary surgeon for any reason.  In Household 7, one cat was treated for FIP and another was suspected of FIP 7 months later, hence he was re-tested to see if his FCoV antibody titre was high or not, in fact it had reduced dramatically, ruling out FIP.  This brought such comfort to his guardians they re- tested the other four cats. 

As laid out in Section 3.2 we had pre- and post-antiviral treatment reducing FCoV antibody titres for a total of 18 cats and failure of 2 new kittens in one household to seroconvert.  Had we included FCoV antibody testing as a criterion for inclusion, there would be no paper due to low numbers.

We could list all the FCoV antibody tests that we have information for, but it would be irrelevant to proving or disproving the hypothesis presented in the paper, and would make the paper even more turgid to read because of all the explanations that would have to then be included. 

Reviewer 2: - Weren’t all cats to be included RT-PCR negative/antibody negative? (or at least reducing antibody levels). What if a cat was RT-PCR negative/serology positive, what happened?

DDA et al reply:  The criteria for inclusion was FCoV RT-PCR faecal test results before and after the antiviral course AND a follow-up period of at least 6 months.  Your last question is interesting to us: we have the impression that most transiently infected cats lose their FCoV antibodies quite quickly (i.e. in under a year) but we don’t have enough data to prove it; whereas the cats who recovered from FIP maintained a high FCoV antibody titre for a considerable time (as we noted in our AGP paper published in 2022).

You can see the fate of some RT-PCR negative/serology positive cats in the comments section of Table 1 for example for Households 1, 2, 5, 6, 7: their antibody titres decline over time and most cats remain well unless they have some other condition.  

Reviewer 2: - Section 3.1. and elsewhere. Is FCoV ‘elimination’ too strong?

DDA et al reply: we wrangled about this before submitting our paper, and in fact the choice of the word “elimination” was a tone down from my initial choice of “eradication” which can apply to a household, but not an individual cat. 

Reviewer 2: - Section 3.1. and elsewhere. Is FCoV ‘elimination’ too strong? What about FCoV localisation elsewhere from the GIT without faecal shedding? My quick reading of other papers is that this can occur? 

DDA et al reply: you’re absolutely correct and possibly thinking of the works of Dr Marina Meli and Prof. Anja Kipar?  You’ll recall that we addressed your second question in line 238 of Discussion (of the previous version): we were also surprised that absolutely no cats seemed to be incubating FIP, or displayed any later evidence of systemic FIP that had not been cleared by the brief antiviral course and we put it down to our relatively small cohort.  We especially worry about virus remaining in the brain since that has been the major bugbear in FIP treatment.  You brought this up again in your review, so we have hugely expanded that section of the Discussion:

“The incubation period for FIP, from infection to the appearance of clinical signs may be a few weeks but can be many months or, rarely, years [31].  In an experimental infection of laboratory cats who were euthanased 80 days after infection FCoV was found in almost all internal organs; the cats had appeared to be healthy apart from transient pyrexia shortly after infection [32]. In a UK study, the virus was found in tissues in 22% of cats without FIP, compared with 95% of cats with FIP [33].  In a field study of the FIP vaccine 31 of 609 cats under one year old developed FIP within 285 days of the placebo or vaccine being administered: viral RNA was detected in stored blood samples taken at the time of vaccination [23]: they were already systemically infected and incubating FIP even though they seemed well.  These studies showed that in cats which appear to be healthy virus can be present systemically, not just in the gastrointestinal tract; therefore we expected that in our group of treated cats there might have been some that were incubating undetected FIP, for whom a short course of antiviral would not be sufficient, since most cats require around 7 to 8 weeks [24] 12 weeks [34, 35] or even—in relapsed cases—two courses [34, 35, 36] of treatment.  However, FIP did not occur in cats whose intestinal FCoV had been eliminated, possibly because of the small size of the cohort of cats.

FCoV can cross the blood-brain barrier, resulting in neurological signs, FIP treatment relapses frequently present with neurological signs [36], therefore we were especially watching for neurological FIP developing.  In a recent study of 26 cats which experienced FIP treatment relapses, 23 were treated using injectable rather than oral GS-441524 [36] but unfortunately no control group of cats without relapses was presented to prove that the reason for the relapses was beginning treatment by injection. In our experience, injectable GS-441524 does not consistently eliminate FCoV from the gut: one cat treated for FIP with GS-441524 injections for 12 weeks still shed virus in his faeces at least two years later [24]: it is from such carrier cats as this that virus mutants resistant to GS-441524 could emerge.”

Our database continues to grow slowly so time will tell, and hopefully people with larger resources than ours will add to the literature.

Reviewer 2: - I couldn’t really follow Table 1 – there was a huge amount of information. Shouldn’t it go later in Results? And is it imperative, or could it be a Supplementary Table to improve the article flow?

DDA et al reply: your feedback was helpful because it made me realise that the table was a monster; so to try to simplify the table a bit, the columns and comments on the 11 non-FIP deaths were moved into a separate Table 2.  We hope this will make the paper easier to read. 

The rule of the journal is to put the table into the manuscript immediately it is first mentioned, therefore we removed the mention of it from Methods, because we agree with you that the table is more appropriately placed in Results. 

We’re reluctant to move the data into Supplementary material because there is less chance that it would be seen there.

Reviewer 2: - Sections 3.3-3.5. I would like to be more convinced that the control group used (reference 15) was a suitable control group. For example follow up in the present study was a period of at least 6 months, where as reference 15 was ‘conducted over a period of 6 years’. What about a statistical comparison of ages, sexes, breed etc. between the 2 groups?

DDA et al reply: I very much regret that much of what you request is no longer possible, due to details of the original data being no longer available, although we’ve now added details about age.  In the 1995 paper the majority of key information was gleaned during the first 36 months post FCoV diagnosis and most FIP death occurred within the first 6 months from FCoV diagnosis.  You’ll understand that it’s not that 820 cats were all found on one calendar date and followed, but that households were identified sequentially and that the data at 6 months often meant that the household was only identified 6 months before the paper was written.   

You asked a question along similar lines below and we have now expanded upon this whole issue hugely in Discussion.

Reviewer 2: - ABSTRACT ‘Cats from 8 households recovered from chronic FCoV enteropathy….’ RESULTS 3.6. ‘Resolution of FCoV-associated chronic enteritis in 8 households’ CONCLUSION ‘The treatment also cured a number of cats that were suffering from chronic gastrointestinal disease.’ I would like a little more evidence for this bold statement about ‘curing’ a chronic disease, to make me believe it. What other work up was performed in these 8 cats, prior to antiviral treatment?

DDA et al reply: admittedly some of these data are anecdotal, as reported by the cat guardians or their veterinarians: we have expanded the description where possible.  The owner of Household 27 furnished me with her histopathology report of the kittens, but I am not at liberty to supply the actual report which contains personal data: the relevant part of the report, i.e. the positive FCoV immunohistochemistry, is presented in Table 1.  We have toned down the sentence in the Conclusion to read:

“The role of FCoV in cats suffering from chronic gastrointestinal disease requires further investigation.”

Please also see our response to your query on this subject below.

Reviewer 2: - And what sort of follow up for each cat?

DDA et al reply: The follow-up is shown in Table 1: the cats reported still well at 26, 27, 36, 19, 18, 16, 15 months, and no more FCoV-related kitten deaths in Household 27 six months and five litters after eradicating the FCoV.

Reviewer 2: DISCUSSION

Reviewer 2: - Any other variables between the current cohort and the historical control group, apart from antiviral treatment, that might have been a factor here? Eg. Different husbandry? Quarantine of shedding animals?

DDA et al reply: another really excellent question.  The biggest variable is that this time we were able to accurately identify FCoV-infected cats and kittens, whereas last time we were inferring infection from FCoV antibody testing, and FIP and FCoV enteritis occurring.  Thus in the present study kittens born into the breeding households post-FCoV-eradication were not counted because we knew with a considerable degree of certainty that they had not been exposed to FCoV: therefore the number of kittens in this present study was lower than in the previous study (the percentage of cat breeders in the present study was also lower).  In the previous study kittens accounted for 22 of the 37 FIP deaths.  The lack of FCoV infection in the kittens born into the catteries in this study would have meant a lower environmental virus load and, as Pedersen & Black showed in 1983, that would have decreased the prevalence of FIP.

In the original paper, the cats were zeroed back to the time of first FCoV or FIP diagnosis in the Kaplan-Meier curve, in this present paper, follow-up was reported from when they became FCoV free. 

You enquired about husbandry: the large Household 8 is a breeding cattery with cats in small groups, but that was also true for some of the cat breeders in the 1995 study.  Another difference could have been that we had published a cat litter study in the interim, but the Dr Elsey Cat Attract litter which fared best in that study for reducing FCoV transmission is not available in the UK and the impact on FIP mortality of using that litter has not been studied.

The start of the Discussion has been massively revised to address this query and we hope that you will find our comparison of the two cohorts satisfactory: there were differences between the two studies, but not such that would detract from the significance of our findings.

Reviewer 2: - L248-250 These were not closed colonies, so how was the risk if re-infection accounted for? Lines like this in the Discussion left me doubting the results and wanting more information on how much testing was performed in order to be certain of FCoV eradication from a household. ‘As expected re-infected cats began to shed virus again and further FIP cases occurred (data not shown).’ Perhaps this data should be shown!

DDA et al reply:  I’ve modified that sentence to read:

“As expected, in that household, re-infected cats began to shed virus again and further FIP cases occurred (data not shown).” to make it clear that we were talking about that single breeding cattery.  We do not have consent to publish those data: that breeder did not want to be a part of this paper. 

Please bear in mind that at least 4 of the 27 households introduced new cats or kittens, did quarantine the newcomers, did faecal or antibody testing, and treating if required, and remained FCoV-free.  The cat breeders tell me they’re doing “spot checks”  i.e. sending faeces from time to time for FCoV RT-PCR testing.  Of course I request copies of their results, but if they’re not forthcoming I have no right to insist.  People are busy and the last three years have been a nightmare for everybody.

Reviewer 2: - L238-240 ‘We expected that in our group of treated cats there might be some that were incubating undetected FIP, for whom a short course of antiviral would not be sufficient, since most cats require around 7-8 weeks [20] or even 12 weeks of treatment [21].’ I think this finding needs to be expanded upon further. Presumably the difference is due to eradicating FCoV from the GIT of healthy cats, versus eradicating systemic FIP from sick cats. But I think for the reader’s benefit this difference needs to be emphasised

DDA et al reply: this is along the lines of your comment above and your comments here were included in our modification and expansion of this section of the Discussion which we showed above.

Reviewer 2: - What is your evidence of ‘early’ elimination of FCoV infection, in a field study without knowledge of day of challenge? ABSTRACT L28, DISCUSSION L219

DDA et al reply: good point - by “early” we mean pre-FIP. There’s no room in the abstract to elaborate, but in Discussion  we now explain this further: “We believe we have demonstrated that early elimination of FCoV infection—by which we mean treating the virus before the cat has developed FIP—is an effective method to prevent FIP.“

Both our 1995 paper and 1992 study of kittens did find that most FIP occurred relatively early in infection.

Reviewer 2: - Some of the writing in this paper I found a little casual and not as sophisticated as I would have expected – e.g.  ‘Making antibodies costs the body energy, and the globulins themselves thicken the blood, therefore the body stops making antibodies as soon as it deems it safe to do so, but keeps memory T cells with the blueprint so it can produce antibodies again if need be.’ Sounds like it has been written for high school biology students to understand rather than intelligent veterinarians and virologists

DDA et al reply: that sentence has now been removed.

Reviewer 2: - L276-277 ‘…response to an oral (not systemic) anti-coronavirus drug should now be included amongst the therapeutic trials performed in suspect feline IBD cases.’ I think this statement needs more qualification. When exactly are the authors advocating for a trial of FCoV antiviral medication? What testing should be performed prior to a treatment trial? What clinical signs (type, duration etc.) would prompt these authors to commence treatment?

DDA et al reply: I’m glad you encouraged us to expand upon this paragraph: 

“The elimination of FCoV cured some cats in the study that were suffering from chronic gastrointestinal conditions (chronic is defined as over three weeks duration [12, 13], and in our cohort some cases had suffered for years).  The presence of FCoV in a diarrhoea infectious disease profile tends to be overlooked because the virus is highly prevalent in purebred cats [8, 26] and multicat environments such as cat shelters [37, 38]. Just as any good gastroenterologist would eliminate Giardia, or Tritrichomonas foetus, or any other known diarrhoea-causing pathogens before conducting a more invasive procedure such as endoscopy or biopsy, the recovery of several cats in our small cohort shows that a response to an oral (not systemic) anti-coronavirus drug should now be included amongst the therapeutic trials performed in suspect FCoV-infected feline IBD cases where the stools remain loose despite elimination of other diarrhoea-causing pathogens, and where there is a lack of response to dietary change and probiotics.”    

We already described which WSAVA IBD criteria FCoV fits in the Introduction, so we have not repeated them in Discussion.

Reviewer 2: What explicitly are these authors suggesting – should all kittens be treated prophylactically with antiviral medication? All kittens/cats rehomed from multi-cat households/shelters? What is the practical outcome of this study?

DDA et al reply: we are not laying down any laws here: this is a scientific paper, not a guideline.  We believe in the freedom of human beings to choose, to make informed decisions and our efforts are to provide information and let people with cats choose for themselves.  Our personal preference would be to reduce virus load in environments, especially those in shelters and breeding catteries and this can be effected by good hygiene, quarantine and testing.  Previously we published the effect of certain cat litters on reducing FCoV transmission and we showed that testing for FCoV antibodies prevented the introduction of the virus into the Falkland Islands.  We are seeking an alternative to GS-441524 to stop FCoV shedding, but with no success so far.  Antiviral products appear to be widely available on the internet and—with few exceptions— the veterinary profession has relinquished responsibility for addressing their use and the public is now going to Facebook group moderators instead for their advice.  This present publication is another attempt to provide accurate information on antivirals in FCoV infection and to discourage irresponsible use of them.

Reviewer 2: REFERENCES

- 22 references (of which 6 references are from the primary author) is underwhelming for an area of veterinary science in which there are many, many publications

DDA et al reply: You are totally correct!  We have a confession to make: we were about to upload the manuscript as a short communication with a reference number limit to another journal, when we got an email from Viruses with an offer we couldn’t refuse,  but that came with a deadline of only 24 hours to reformat our paper.  In my defence about the number of references which are my own – there are not many people doing this kind of long term follow up of naturally infected cats for me to cite.   The plethora of data on experimental infections of SPF cats is largely not helpful in this study, and has often given results in conflict with what occurs in natural infections in the real world.

We’re sorry you opted to not sign your review report, because your contribution made an enormous improvement to our paper.  God bless you and your cats.

Reviewer 3 Report

It is an interesting story for Stopping feline coronavirus shedding prevented feline infectious peritonitis.

 The authors should afford the permission for GS-441524 Permissions in cats.

Author Response

Reviewer 3: It is an interesting story for Stopping feline coronavirus shedding prevented feline infectious peritonitis.

DDA et al reply:  Thank you very much for describing our study as interesting and for the time you spent reading and reviewing it.  We hope we have improved our paper to your satisfaction?

Your review seemed very short, but MDPI assure me that nothing was missing from the website, so please let MDPI know if we appear to not have answered points which you brought up.

Reviewer 3: The authors should afford the permission for GS-441524 Permissions in cats.

DDA et al reply: we’re sorry, we don’t understand this sentence.  In the UK, we are fortunate in having GS-441524 available from Bova Specials, which is a compounding pharmacy with  permission from our Veterinary Medicine Directorate to make drugs for veterinary use.  Bova are also negotiating with the regulatory bodies of other countries to make the drug available (though sadly it is still unaffordable for many cat people).   I hope that covers what you asked, but if not, please ask again.

Round 2

Reviewer 2 Report

I have nothing further to add!

All of my questions/queries have been comprehensively addressed by the authors, thank you. In particular I commend the authors on a beautifully reworked Discussion, and so many nuggets of wisdom that will make this paper a very important contribution to the field (e.g. "We believe that we have demonstrated that early elimination of FCoV infection—by which we mean treating the virus before the cat has developed FIP—is an effective method to prevent FIP" and "it is essential that a post-treatment faecal RT-PCR test should be performed to ensure that virus elimination has been effected" - in my limited experience the latter is not always happening).

If I could please feed back to the authors how much I enjoyed reading their work and in particular their responses to my questions. Not only were their responses informative, patient and well explained, but they were so warm and respectful towards my review. I have learned so much from diving into this article and I thoroughly enjoyed the review process and interacting with this group.

Well done again on an outstanding effort and an important (and ongoing) contribution to FIP, virology, and feline medicine in general :-)

(PS I am sorry not to personally sign off my report but I am not confident enough to do that!!!)